# Micronutrient intake and prevalence of micronutrient inadequacy among women (15-49 y) and children (6-59 mo) in South Kivu and Kongo Central, Democratic Republic of the Congo (DRC)

Najma A. Moumin[1,2☯], Moira Donahue Angel[3☯], Crystal D. Karakochuk[4], Kristina D. Michaux[4], Mourad Moursi[3], Kossiwavi Améwono Ayassou Sawadogo[5], Jennifer Foley[3], Meaghan D. Hawes[4], Kyly C. Whitfield[6], Pierrot L. Tugirimana[7,8], Esto Bahizire[9,10,11], Pierre Z. Akilimali[12], Erick Boy[3], Thomas R. Sullivan[1,13], Tim J. Green[1,2,4]*

1 Division of Healthy Mothers, Babies & Children, South Australian Health and Medical Research Institute, Adelaide, SA, Australia, 2 Discipline of Paediatrics, University of Adelaide, Adelaide, Australia, 3 HarvestPlus c/o International Food Policy Research Institute, Washington, DC, United States of America, 4 Food, Nutrition and Health, University of British Columbia, Vancouver, BC, Canada, 5 Initiatives Conseil International–Santé, Ouagadougou, Burkina Faso, 6 Department of Applied Human Nutrition, Mount Saint Vincent University, Halifax, NS, Canada, 7 Faculty of Medicine, University of Goma, Goma, Democratic Republic of the Congo, 8 Department of Clinical Biology, College of Medicine and Health Science, University of Rwanda, Kigali, Rwanda, 9 Faculty of Medicine, Université Catholique de Bukavu, Bukavu, Democratic Republic of the Congo, 10 Center of Research in Natural Sciences of Lwiro, Bukavu, Democratic Republic of the Congo, 11 Center of Research in Epidemiology, Biostatistics and Clinical Research, Université Libre de Bruxelles, Brussels, Belgium, 12 Department of Nutrition, Kinshasa School of Public Health, University of Kinshasa, Kinshasa, Democratic Republic of the Congo, 13 School of Public Health, University of Adelaide, Adelaide, Australia

☯ These authors contributed equally to this work.
* tim.green@sahmri.com

## Abstract

Iron biofortified beans and carotenoid enriched cassava are proposed as a solution to combat iron and vitamin A deficiencies, respectively, in the Democratic Republic of Congo (DRC). To inform the need for biofortified foods, we conducted a survey in 2014 in two provinces of the DRC, South Kivu and Kongo Central. Unexpectedly, women of reproductive age (WRA; 15–49 y) and their children (6–59 m) had a low prevalence of biochemical iron and vitamin A deficiency, based on ferritin and retinol binding protein, respectively. To better understand the lack of biochemical deficiency of these nutrients, we examined the prevalence of inadequate intake for these and other select nutrients. Dietary intake was assessed using 24-hour recalls among 744 mother-child dyads. Repeat recalls on a non-consecutive day were conducted with a subsample of the study population to account for intra-individual variation and estimate usual intake. In WRA, the prevalence of inadequate iron intakes were 33% and 29% in South Kivu and Kongo Central, respecitvely. The prevalence of inadequate vitamin A intakes among WRA was low in South Kivu (18%) and negligible in Kongo Central (1%). Iron inadequacy was highest in infants (6–11 m) at 82% and 64% in South Kivu and

**Data Availability Statement:** Data files can be accessed at the following DOI link (https://doi.org/10.7910/DVN/RNWYR8).

**Funding:** The funds for the study were made available from the Bill and Melinda Gates Foundation to HarvestPlus through Grant No. OPP1019962, and funds were made available from HarvestPlus to UBC through agreement 2014H8307. MDA, MM, JF and EB are HarvestPlus employees and were involved in the design, collection, analysis and/or preparation of the manuscript. TG had primary responsibility for the final content.

**Competing interests:** The authors have declared that no competing interests exist.

Kongo Central, respectively. Among older children (12–59 m) in both provinces, the prevalence of iron inadequacy was similar at ~20%. There was a high prevalence of inadequate zinc intake in women and children (i.e. 79–86% among WRA and 56–91% among children 6–59 m) consistent with our findings of a high prevalence of low serum zinc in the same sample. Dietary data here corroborate the low prevalence of biochemical vitamin A deficiency but not iron. However, any change to the supply of red palm oil (primary source of vitamin A) would dramatically reduce population vitamin A intakes, thus a carotenoid enriched cassava program may be beneficial as a safety net measure. Crops biofortified with zinc also appear warranted. We caution that our findings cannot be extrapolated to the entire Congo where diverse agro-ecological landscape exist or when political and environmental shocks occur which challenge food production.

## Introduction

Micronutrient deficiencies leading to anemia are thought to be common in the Democratic Republic of the Congo (DRC), especially among children and women of reproductive age (WRA). According to the most recent population survey (2013–2014), the overall prevalence of anemia was ~35% among WRA (15–49 years) and ~60% in children (6–59 months) [1]. Anemia has been mainly attributed to iron deficiency, but may also be linked to other micronutrient deficiencies including folate, vitamins A and B$_{12}$ [2], and zinc [3]. Unfortunately, very little is known about the micronutrient status of the Congolese. HarvestPlus has plans to scale up their program on iron biofortified beans and carotenoid biofortified cassava in the DRC to help control anemia as well as reduce blindness and decrease mortality from infectious disease caused by vitamin A deficiency.

To inform the need for the intervention we conducted a micronutrient survey of 744 mother-child pairs in two provinces in the DRC, South Kivu and Kongo Central. Despite relatively high rates of anemia, we found very low rates of biochemical iron (low ferritin) or vitamin A deficiency (low retinol binding protein) [4]. Iron deficiency anemia (low ferritin with anemia) was present in less than 1% of WRA and affected about 4% of children overall, rising to 9% in children 6–23 months in South Kivu. We found very little evidence of biochemical vitamin B$_{12}$ or folate deficiency but found low serum zinc in 50% of WRA and around 25% of young children [4].

We are aware that there are other causes of anemia such as malaria, helminths infection, and the presence of genetic blood disorders [5–7]. Nevertheless, we were perplexed by the general lack of micronutrient deficiencies in this population and sought corroborating evidence from dietary intake information that was obtained from 24 hour recalls in the study group. Here we aim to estimate the prevalence of inadequacy for select nutrients among WRA (15–49 y) and children (6–59 m), to identify major dietary sources of nutrients, and to quantify the intakes of potential vehicles (i.e. bean) to assess the potential impact of biofortifcation.

## Materials and methods

### Study design

A cross-sectional survey using the multiple pass 24-hour dietary recall (24HDR) interview method was conducted from August to October 2014 to assess usual dietary intake and estimate the prevalence of inadequacy for select nutrients in two contrasting provinces in the

DRC, South Kivu and Kongo Central. This was part of a larger nutritional assessment survey in the two provinces, the biochemical results of which have been reported on in Harvey-Lee-son et al. [4] These two provinces were selected as they were politically stable at the time of data collection and captured the country's diverse geographical landscape. The densely populated eastern province of South Kivu was selected for its high altitude, where most of the population lives approximately 1000 m above sea level. In contrast, Kongo Central, the westernmost province, traversed by the Congo River, was selected as it was located in a lowland area. Ethical approval was obtained from the Clinical Research Ethics Board at the University of British Columbia (H14-01279), the Université de Kinshasa (ESP/CE/033/14), and the Université Catholique de Bukavu (UCB/CIENC/25/2014).

## Participants

The sampling method used for this study has been described in detail elsewhere [4]. Briefly, participants were selected using a three-stage probability proportionate to size (PPS) sampling method. During the first stage, only those health zones in closest proximity to biological sample processing centres in Bukavu and Kinshasa were included in the sampling frame to maintain the integrity of the blood samples collected concurrently with the 24 HDRs. In South Kivu, 10 health zones within a 60-km radius of Bukavu were selected, and in Kongo Central all six health zones within the district of Lukaya (nearest to Kinshasa) were selected. Villages were then randomly selected proportionate to population size (South Kivu: n = 40 villages; Kongo Central: n = 25 villages). Finally, households were randomly selected using the random walk method; three enumerators walked in opposite directions from one another at 120˚ starting from the geographical centre of each village. Every tenth household that met inclusion criteria was then selected and enrolled into the study until 12 households were selected from each village. If the household did not meet criteria or refused to participate, the next available household in sequence that met eligibility criteria was chosen.

Households were included in the cross-sectional study if participating mothers met the following criteria: 1) mothers were the female heads of the selected household; 2) were non-pregnant and between 15–49 years of age; 3) had no serious health conditions or chronic diseases; 4) had a child 6–59 months of age living in the home; and 5) agreed to provide consent to draw blood samples for themselves and their child.

The sample sizes in this study were calculated to measure the expected point estimates of the prevalence of anemia among children 6–59 months of age in the two study provinces. According to the most recent national demographic and health survey, the prevalence of anemia among this age group was 60% in South Kivu and 71% in Kongo Central [1]. With a margin of error of ±7.5%, and a 95% confidence interval with alpha 0.05, we estimated a minimum sample size of 328 and 287 mother-child pairs in South Kivu and Kongo Central respectively would be needed. The number of villages (clusters) and units of observation were determined in keeping with micronutrient survey sampling guidelines and adjusted proportionate to population size [8]. In total, n = 40 and n = 25 clusters with 12 households per cluster, in South Kivu and Kongo Central, respectively were included. However, due to unexpected security issues in one health zone, the final sample size in South Kivu was reduced to n = 444 mother-child pairs. In Kongo Central, the final sample size was n = 300 mother-child pairs.

## Data collection

After eligibility was confirmed, women heads of selected households completed a baseline questionnaire designed to elicit information on household demographics, as well as current knowledge and practices regarding health, nutrition, and water, sanitation and hygiene

(WASH) behaviours. Following this questionnaire, anthropometric measurements (height/length, weight and mid upper arm circumference [(MUAC) children only]) were taken, along with non-fasting blood samples for each mother-child pair. The baseline questionniare was developed in English, translated into French and pilot tested in non-study participants in each of South Kivu and Kongo Central. Revisions were made after pre-testing to ensure translations were site-specific. Finally, a multi-pass 24 HDR interview was completed with all participants, the results of which are presented in this paper.

**Dietary assessment.** The 24 HDR is a validated and widely used data collection instrument that helps gather detailed information on all foods and beverages consumed by a respondent in the preceding 24 hours [9, 10] This method of dietary assessment is particularly ideal for use in low resource settings as it is inexpensive, uses little equipment, and takes on average less than one hour to complete. Despite its ease of use however, there are limitations including: over/under estimation of food and beverages consumed; poor estimation of portion sizes; and failure to capture day-to-day variations in food intake. Gibson and Ferguson addressed these limitations and subsequently developed the multi-pass 24 HDR [11]. For our study, an adapted version of the multi-pass 24 HDR method, tailored for use in the DRC context, was used.

**Food composition table development.** To develop a locally appropriate food composition table and means of determining portion size, focus group discussions were held with women in communities neighbouring the study villages. Three focus groups were held in three non-study villages in each of South Kivu and Kongo Central each consisting of 10–14 purposefully selected women. A local female project officer conducted the focus groups with women at their respective village health centers in line with protocols described in Gibson and Ferguson [11]. Focus group discussions continued until the point of saturation was reached, that is, consensus on commonly consumed local dishes and their composition. From these discussions, information on common and unique household recipes was systematically collected and used to develop a recipe database. Trained enumerators collected household recipes identified in the focus groups from purposefully selected respondents. Edible portions of raw ingredients were recorded and the cooked weight of each mixed dish was measured after the cooking process. Recipes were then standardized to predetermined cooked portions in the recipe database. This information was used to generate conversion factors (factors to allow reported quantities of food to be converted to grams, including volume equivalents, household measures, clay models, and linear dimensions) and a nutrient composition table of local foods [12]. A nutrient composition table from a HarvestPlus study in Uganda [13] was used as the starting point and then completed with the USDA Nutrient Database 22 [14]. Raw foods were used and USDA retention factors applied to account for water and nutrient losses due to cooking. Because red palm oil is an important source of provitamin A carotenoids (pVACS) in the DRC, we conducted a post hoc sampling of red palm oil from street side vendors in nine distinct markets located in South Kivu, Kongo Central and three additional provinces in the DRC and analysed the samples for carotenoids, which had a mean of 400 μg/g pVAC (range: 273–600 μg/g pVAC). The average of the results of was then applied to the food composition table, assuming a bioconversion factor of 7 to 1 micrograms of beta carotene equivalents to retinol and a retention of 70% from cooking [15] Nutrient intakes from breastmilk were also estimated for partially breastfed children based on the World Health Organization (WHO) average daily intake scenario based on age category [16]. Finally, we created new variables for cassava and beans to express the reported intakes as dry weight beans and fresh weight peeled cassava, by applying conversion factors to the most commonly reported foods. Gram weight of cooked beans was divided by 2.54 to derive dry weight bean consumption, and uncooked cassava flour used for *fufu* was divided by 0.31 and uncooked fermented dough used for

*chikwangwe* by 0.23 to derive gram weight of peeled cassava roots. Nearly all the maize consumed in the 24HDR was reported as maize flour, thus no conversion factors were applied.

**Structured interviews.**   A pre-interview visit from trained enumerators was conducted two days before each 24 HDR. Respondents were presented with a picture chart of locally appropriate foods and beverages and asked to tick all items consumed by themselves and their child for the following day. In addition, respondents were asked to save all packaging from commercial foods and beverages and use standardized plates, cups, bowls and utensils to help quantify their consumption throughout the following day. On the interview day, respondents were asked to produce the picture chart and all food packages from the previous day. The interview was then conducted in four separate sections and information on the quantity, type, and method of preparation for each food item was recorded. Probing techniques were used to obtain detailed descriptions of foods, and visual aids (playdough and rice) were used to assist with estimation of portion sizes. Repeat recalls were completed on non-consecutive days with a subset of the sample, ~20% of participants, to account for day-to-day variation in intake. All interviews were conducted with the mother; if another caregiver fed the child, they were interviewed for the child's recall. Mothers were asked about their own food and beverage consumption first, followed by their child. All data collection took place between August and October 2014.

Data management and analysis. Trained enumerators using established guidelines recorded all interview data on hard-copy questionnaires [11]. Field supervisors reviewed each questionnaire for accuracy and completion before data entry into CSDietary software [17] Using this software, macronutrient (energy) and micronutrient (iron, zinc, vitamin C, thiamine, riboflavin, niacin, vitamin B6, folate, vitamin $B_{12}$, and calcium) intakes were quantified. Within-person variability in intake was adjusted for by implementing the ISU method [18] with the Intake Modelling, Assessment and Planning Program (IMAPP) software [19] For foods and nutrients that were not consumed nearly every day, we used the PC Software for Intake Distribution Estimation (PC-Side) program which estimates the joint distribution of usual intake and probability of consumption [20]. The estimated average requirement (EAR) cut point method developed by the Institute of Medicine (IOM) was used to estimate prevalence of inadequate intake for all micronutrients except iron, for which the probability density approach was used due to the skewed distribution of requirements in women and children [21]. The probability of inadequacy was calculated by comparing the sample's iron distribution to the iron requirement distribution percentiles at 7.5% and 10% bioavailability [11] For infants, we adopted the iron intake variation components from among the 1 to 3 year olds to estimate prevalence of inadequacy since the sample size and number of repeated recalls for infants was too small to estimate these parameters. The Institute of Medicine EAR values were used for all nutrients, except for zinc [21]. For zinc, we used the zinc physiological requirements from the European Food Safety Authority [22] and applied bioavailability estimates for an unrefined cereal diet from IZINCG to calculate an EAR appropriate for this setting. These were determined to be the most up-to-date evidence on zinc requirements, given the marked differences in zinc physiological requirements that are reported by various expert groups [23, 24]. Continuous characteristics using unequal variances t-tests and categorical characteristics using Fisher exact tests were completed to compare differences in micronutrient intake and prevalence of inadequacy between provinces. Data analysis was conducted using Stata version 15 [25].

## Results

The response rates for the dietary surveys were 87% in South Kivu and 97% in Kongo Central. The participation rate in South Kivu was lower due to security concerns as one of the villages

was the site of a rebel conflict and was therefore withdrawn from data collection. No mother/child dyad was excluded due to the presence of chronic disease. Households were slightly larger in South Kivu with 6.6 ± 0.12 members and had a greater proportion of women, 48%, reporting no formal schooling. Kongo Central, on the other hand, had smaller household sizes with 5.7 ± 0.13 members and had a higher proportion of women, 43%, reporting completion of secondary school (Table 1). Access to health services were similar between provinces. Approximately 90% of all children received vitamin A capsules, and roughly 80% were treated with de-worming tablets in the past six months. Iron supplementation was less common with less than 10% of children receiving tablets or syrup in the preceding six months.

## Mean nutrient intake and prevalence of inadequacy

The mean daily nutrient intakes and prevalence of inadequacy among WRA in South Kivu and Kongo Central are summarized in Table 2, and in Tables 3–5 for their children 6–59 months of age. Mean ± SE vitamin A intakes were lower among WRA in South Kivu than Kongo Central (757 ± 14 µg vs. 1109 ± 19 µg RAE/d, p<0.001). The prevalence of Vitamin A inadequacy among WRA was moderate in South Kivu (18%) and absent in Kongo Central (1%). Iron intake was slightly lower among women in South Kivu compared to Kongo Central (18.8 ± 0.3 mg/d vs 20.3 ± 0.5 mg/d, p = 0.01), respectively. Assuming a mixed vegetarian diet and moderate bioavailability of 10%, the prevalence of inadequacy for iron was similar in both provinces at ~30%. The prevalence of inadequate zinc intakes was high in both provinces at 86% in South Kivu and 79% Kongo Central. In both provinces, the prevalence of inadequacy was high for thiamine, riboflavin, and calcium, but moderate for folate, niacin, and vitamin $B_{12}$.

Among children, age-specific EARs, IZINCG values, or iron distribution percentiles were applied. The prevalence of inadequacy for infants 6–11 months was calculated for iron and zinc only because other nutrients lack an EAR (Table 3). The prevalence of zinc inadequacy was high among infants, albeit significantly higher in South Kivu than Kongo Central (91% vs 67%, p<0.005). Similarly, a higher proportion of infants in South Kivu (82%) had inadequate iron intakes at 10% bioavaialability compared to Kongo Central (64%).

**Table 1. Household characteristics of the study populations South Kivu and Kongo Central, Democratic Republic of the Congo[a].**

|  | South Kivu | Kongo Central |
| --- | --- | --- |
|  | N = 444 | N = 300 |
| Household size, mean ± SE | 6.6 ± 0.12 | 5.7 ± 0.13 |
| Mother's age, y, mean ± SE | 29.2 ± 0.02 | 29.9 ± 0.02 |
| Lactating Mothers | 386 (87) | 219 (73) |
| Mother's education level |  |  |
| -No schooling | 214 (48) | 29 (10) |
| -Primary school | 139 (31) | 141 (47) |
| -Secondary school | 87 (20) | 128 (43) |
| -Post-Secondary | 4 (1) | 1 (0) |
| Child currently breastfeeding | 322 (73) | 200 (67) |
| Child received vitamin A capsule in past 6 mo | 397 (89) | 279 (93) |
| Child received de-worming in past 6 mo | 368 (83) | 232 (77) |
| Child received oral iron in past 6 mo | 36 (8) | 30 (10) |

[a]Values are n (%) unless otherwise stated.

**Table 2. Mean daily energy and micronutrient intakes of WRA and prevalence of inadequacy.**

| | EAR | South Kivu (n = 441) | | Kongo Central (n = 290) | | | |
| --- | --- | --- | --- | --- | --- | --- | --- |
| | | Intake [a] | Prevalence of Inadequacy[b] | Intake | Prevalence of Inadequacy[b] | P-value (Intake) | P-value (Prevalence of Inadequacy) |
| Energy, kcal | -- | 2372 ± 40 | | 2338 ± 48 | | 0.59 | -- |
| Vitamin A, µg RAE[c] | 500 | 757 ± 14 | 18 ± 8 | 1109 ± 19 | 1 ± 5 | <0.001 | <0.001 |
| Iron, mg | 8.1 | 18.8 ± 0.3 | | 20.3 ± 0.5 | | 0.01 | |
| 10% bioavailability | | | 33[d] | | 29[d] | | |
| 7.5% bioavailability | | | 58[d] | | 52[d] | | |
| Zinc, mg | 11.6 | 8.3 ± 0.2 | 86 ± 5 | 9.0 ± 0.3 | 79 ± 4 | 0.05 | 0.02 |
| Vitamin C, mg | 60 | 83 ± 1 | 22 ± 9 | 90 ± 1 | 8 ± 24 | <0.001 | <0.001 |
| Thiamine, mg | 0.9 | 0.95 ± 0.02 | 53 ± 3 | 1.0 ± 0.02 | 43 ± 3 | 0.08 | 0.01 |
| Riboflavin, mg | 0.9 | 0.98 ± 0.02 | 47 ± 3 | 1.0 ± 0.03 | 46 ± 4 | 0.58 | 0.82 |
| Niacin, mg | 11 | 14.4 ± 0.2 | 26 ± 5 | 13.7 ± 0.3 | 35 ± 4 | 0.05 | 0.01 |
| Vitamin B6, mg | 1.1 | 2.1 ± 0.03 | 4 ± 4 | 1.6 ± 0.04 | 20 ± 5 | <0.001 | <0.001 |
| Folate, µg | 320 | 522 ± 13 | 21 ± 9 | 508 ± 15 | 24 ± 5 | 0.48 | 0.36 |
| Vitamin B12, µg | 2 | 14.5 ± 12 | <1[e] | 2.0 ± 0.2[f] | NE[g] | 0.3 | -- |
| Calcium, mg | 800 | 581 ± 11 | 85 ± 6 | 789 ± 20 | 59 ± 5 | <0.001 | <0.001 |

[a]Intake data are presented as mean ± SEM and represent the usual nutrient intake for women in South Kivu and in Kongo Central.

[b]Prevalence of inadequacy was estimated as the percent of the usual intake distribution below the estimated average requirement (EAR) for each micronutrient (IOM, 2000) except for iron. For iron, the full probability approach at 7.5% and 10% bioavailability was used (Gibson and Ferguson, 2008). Estimated average requirement for zinc was calculated using physiologic requirements reported by EFSA (EFSA, 2014) and bioavailability assumptions for unrefined cereal diets (IZINCG, 2004). The Institute of Medicine's estimated average requirements for all other nutrients were used. Standard errors (SE) are estimated and reported by IMAPP for the cut-point approach and are reported here.

[c]As retinol activity equivalents (RAE). 1 RAE = 1µg retinol, 12 µg β-carotene, 24 µg α-carotene, or 24 µg β-cryptoxanthin. The RAE for dietary provitamin A carotenoids is two-fold greater than retinol equivalents (RE), whereas the RAE for preformed vitamin A is the same as RE.

[d]Robust SEM could not be calculated as the probability approach rather than the EAR cut-point method was used.

[e]PC-Side was used to estimate B12 because it appeared episodically in the diet; the software does not report SEs for the prevalence of inadequacy estimates.

[f]Usual intake could not be estimated thus mean ± SEM of a single day's intake for each woman are presented.

[g]Not estimable. Distribution was highly skewed.

Results for children 1–3 y and 4–6 y of age are summarized in Tables 4 and 5, respectively. Similar patterns observed with maternal vitamin A, iron, and zinc intakes were found among children. Among both age groups, mean vitamin A intakes were lower in South Kivu than in Kongo Central (497± 12 vs 656 ± 18, p<0.001 among 1-3y and 503± 25 vs 814± 89, p = 0.002 among 4-6y); however, the prevalence of inadequate vitamin A intakes was less than 10% in both provinces and age groups. The probability of inadequate iron intakes (10% bioavailability) was much lower among children than their mothers at only 20–25%. In contrast, zinc inadequacy was equally as high, although slightly higher among children 4–6 y compared to 1–3 y at 69% and 57%, respectively. Calcium inadequacy was quite pronounced among all children, 76% among 1–3 years of age and 95% among the 4–6 year old age group. For other nutrients the prevalence of inadequacy was similar to their mothers.

**Food sources of nutrients.** The major food sources contributing to iron and vitamin A were investigated due to the unexpected lack of biochemical deficiency for both nutrients. In contrast, food sources contributing to zinc were assessed due to both the high burden of biochemical deficiency and dietary inadequacy. Results for WRA are presented in Table 6. In South Kivu, the three largest contributors to both iron and zinc intake were cassava flour, beans, and fish. Together these three sources accounted for two thirds of all iron intake and

**Table 3. Mean daily energy and micronutrient intakes of infants 6–11 mo. and prevalence of inadequacy.**

| | EAR | South Kivu (n = 441) | | Kongo Central (n = 290) | | P-value (Intake) | P-value (Prevalence of Inadequacy) |
| | | Intake [a] | Prevalence of Inadequacy[b] | Intake | Prevalence of Inadequacy[b] | | |
|---|---|---|---|---|---|---|---|
| Energy, kcal | -- | 942 ± 35 | | 1040 ± 40 | | 0.07 | |
| Vitamin A, µg RAE[c] | -- | 349 ± 16 | | 641 ± 43 | | <0.001 | |
| Iron, mg | 6.9 | 4.3 ± 0.3 | | 6.4 ± 0.6 | | 0.003 | |
| 10% bioavailability | | | 82[d] | | 64[d] | | |
| 7.5% bioavailability | | | 92[d] | | 76[d] | | |
| Zinc, mg | 3.5 | 2.3 ± 0.1 | 91 ± 18 | 3.1 ± 0.2 | 67 ± 15 | <0.001 | 0.005 |
| Vitamin C, mg | -- | 45 ± 1 | -- | 48 ± 2 | | 0.19 | |
| Thiamine, mg | -- | 0.33 ± 0.02 | -- | 0.42 ± 0.02 | | 0.002 | |
| Riboflavin, mg | -- | 0.47 ± 0.02 | -- | 0.51 ± 0.02 | | 0.16 | |
| Niacin, mg | -- | 5.1 ± 0.3 | -- | 5.1 ± 0.3 | | 1 | |
| Vitamin B6, mg | -- | 0.64 ± 0.04 | -- | 0.57 ± 0.04 | | 0.22 | |
| Folate, µg | -- | 134 ± 4 | -- | 152 ± 8 | | 0.05 | |
| Vitamin B$_{12}$, µg | -- | 5.6 ± 1.2 | -- | 1.2 ± 0.1 | | <0.001 | |
| Calcium, mg | -- | 309 ± 6 | -- | 449 ± 14 | | <0.001 | |

[a]Intake data are presented as mean ± SEM and represent the usual nutrient intake for infants in South Kivu and Kongo Central; the SEM refers to inter-individual variation.

[b]Prevalence of inadequacy was estimated as the percent of the usual intake distribution below the estimated average requirement (EAR) for each micronutrient (IOM, 2000) except iron and zinc. For iron, the full probability approach at 7.5% and 10% bioavailability was used (Gibson and Ferguson, 2008), and for zinc, IZINCG physiologic requirements were used (EFSA, 2014). Standard errors (SE) are estimated and reported by IMAPP for the cut-point approach and are reported here.

[c]As retinol activity equivalents (RAE). 1 RAE = 1µg retinol, 12 µg β-carotene, 24 µg α-carotene, or 24 µg β-cryptoxanthin. The RAE for dietary provitamin A carotenoids is two-fold greater than retinol equivalents (RE), whereas the RAE for preformed vitamin A is the same as RE.

[d]Robust SEM could not be calculated as the probability approach rather than the EAR cut-point method was used.

over half of all zinc intake. Women in this province also consumed more red meat than women in Kongo Central; beef consumption accounted for 6% of zinc intake. Among WRA in Kongo Central, plant-based sources, specifically, cassava flour, amaranth/bean/cassava leaves, and beans provided most of the iron and zinc. As for vitamin A, similar trends were observed in both provinces; red palm oil provided ~70% of the vitamin A among women.

For children in both provinces, the top three food sources for iron and zinc were similar to their mothers (Table 7). In both provinces, cassava flour contributed to approximately 30% and 20% of dietary iron and zinc, respectively. Like their mothers, animal source food consumption was higher among children in South Kivu, where fish and beef contributed to ~20% of dietary zinc. Red palm oil also accounted for 70% of total vitamin A intake among children in both provinces

Staple food intakes for mothers and children are reported in Table 8. Beans, expressed as dry weight, were consumed regularly in both regions, although not every day. On days when women reported consuming beans, ~240 g/d were consumed in South Kivu and 180 g/d in Kongo Central. On average, mean bean intake among women was 93 ± 7 g/d in South Kivu and 59 ± 11 g/d in Kongo Central. Children also consumed a sizeable amount of beans on days of consumption (~130 g/d in South Kivu and 100 g/d in Kongo Central); however, usual intake on all days averaged out to 47 ± 2 g/d in South Kivu and 37 ± 4 g/d in Kongo Central. Cassava intake, expressed as fresh weight peeled cubes, was consumed in large quantities at ~1260 g/d among women in South Kivu and ~1480 g/d in Kongo Central. Cassava consumption among

**Table 4. Mean daily energy and micronutrient intakes of children 1–3 y and prevalence of inadequacy.**

| | EAR | South Kivu (n = 441) | | Kongo Central (n = 290) | | P-value (Intake) | P-value (Prevalence of Inadequacy) |
| --- | --- | --- | --- | --- | --- | --- | --- |
| | | Intake [a] | Prevalence of Inadequacy[b] | Intake | Prevalence of Inadequacy[b] | | |
| Energy, kcal | -- | 1343 ± 29 | | 1190 ± 13 | | <0.001 | 0.002 |
| Vitamin A, µg RAE[c] | 210 | 497 ± 12 | 6 ± 5 | 656 ± 18 | 1 ± 3 | <0.001 | |
| Iron, mg | 3.0 | 10.0 ± 0.3 | | 10.0 ± 0.3 | | 1 | |
| 10% bioavailability | | | 23[d] | | 21[d] | | |
| 7.5% bioavailability | | | 37[d] | | 35[d] | | |
| Zinc, mg | 4.7 | 4.8 ± 0.1 | 56 ± 3 | 4.7 ± 0.1 | 58 ± 4 | 0.48 | 0.66 |
| Vitamin C, mg | 13 | 57 ± 1 | <1 ± <1 | 52 ± 1 | <1 ± <1 | <0.001 | 1 |
| Thiamine, mg | 0.4 | 0.56 ± 0.01 | 31 ± 4 | 0.59 ± 0.01 | 15 ± 8 | 0.03 | <0.001 |
| Riboflavin, mg | 0.4 | 0.61 ± 0.01 | 18 ± 6 | 0.66 ± 0.02 | 14 ± 6 | 0.03 | 0.3 |
| Niacin, mg | 5 | 8.3 ± 0.2 | 16 ± 5 | 7.1 ± 0.2 | 21 ± 5 | <0.001 | 0.14 |
| Vitamin B6, mg | 0.4 | 1.2 ± 0.03 | 1 ± 2 | 0.81 ± 0.02 | 5 ± 4 | <0.001 | 0.004 |
| Folate, µg | 120 | 315 ± 10 | 6 ± 5 | 288 ± 6 | <1 ± 3 | 0.02 | 0.002 |
| Vitamin $B_{12}$, µg | 0.7 | 7.8 ± 3.1 | 6[e] | 1.1 ± 0.03 | <1[e] | 0.03 | -- |
| Calcium, mg | 500 | 363 ± 8 | 84 ± 6 | 466 ± 13 | 65 ± 7 | <0.001 | <0.001 |

[a]Intake data are presented as mean ± SEM and represent the usual nutrient intake of children 1–3 years old; the SEM refers to inter-individual variation.

[b]Prevalence of inadequacy was estimated as the percent of the usual intake distribution below the estimated average requirement (EAR) for each micronutrient (IOM, 2000) except for iron. For iron, the full probability approach at 7.5% and 10% bioavailability was used (Gibson and Ferguson, 2008). Estimated average requirement for zinc was calculated using physiologic requirements reported by EFSA (EFSA, 2014) and bioavailability assumptions for unrefined cereal diets (IZINCG, 2004). The Institute of Medicine's estimated average requirements for all other nutrients were used. Standard errors (SE) are estimated and reported by IMAPP for the cut-point approach and are reported here.

[c]As retinol activity equivalents (RAE). 1 RAE = 1µg retinol, 12 µg β-carotene, 24 µg α-carotene, or 24 µg β-cryptoxanthin. The RAE for dietary provitamin A carotenoids is two-fold greater than retinol equivalents (RE), whereas the RAE for preformed vitamin A is the same as RE.

[d]Robust SEM could not be calculated as the probability approach rather than the EAR cut-point method was used.

[e]PC-Side was used to estimate $B_{12}$ because it appeared episodically in the diet; the software does not report SEs for the prevalence of inadequacy estimates.

children was also high at ~ 450 g/d and 640 g/d in Kongo Central. Maize was consumed infrequently by half the population in South Kivu and a third of the population in Kongo Central at the time of this study, with an estimated mean intake of 20 ± 5 g/d among women in South Kivu and 4 ± 1 g/d intake in Kongo Central. Usual mean intakes of maize among children were similarly very low.

## Discussion

In this paper, we present the results on dietary intakes and prevalence of nutrient inadequacy from our 24 HDR survey to better understand the lack of biochemical deficiency among women and children in South Kivu and Kongo Central in the DRC. To our knowledge, this is the only assessment of dietary intake and micronutrient status among women of childbearing age and their children 6–59 months in these two provinces. When comparing nutrient intakes at the regional level, respondents in Kongo Central had higher mean intakes for most nutrients and consequently lower prevalence of inadequacy. Kongo Central is located in proximity to Kinshasa, the nation's capital, has more commercial activity than South Kivu and is a large supplier of cassava. South Kivu, on the other hand, borders Rwanda and Burundi, thus we expected similar dietary patterns to these bordering countries characterized by more bean and maize consumption. Given that South Kivu is situated within Lake Kivu and Lake Tanganyika, we also expected respondents in this region to have significantly higher fish consumption than

**Table 5. Mean daily energy and micronutrient intakes of children 4–6 y and prevalence of adequacy.**

| | EAR | South Kivu (n = 56) | | Kongo Central (n = 29) | | P-value (intake) | P-value (prevalence of Inadequacy) |
| | | Intake[a] | Prevalence of Inadequacy[b] | Intake[a] | Prevalence of Inadequacy[b] | | |
|---|---|---|---|---|---|---|---|
| Energy, kcal | | 1376 ± 47 | | 1463 ± 88 | | 0.39 | |
| Vitamin A, µg RAE[c] | 275 | 503 ± 25 | 10 ±13 | 814 ± 89 | 7 ± 8 | 0.002 | 0.71 |
| Iron, mg | 4.1 | 11.4 ± 2.0 | | 13.2 ± 1.5 | | 0.47 | |
| 10% bioavailability | | | 24[d] | | 24[d] | | |
| 7.5% bioavailability | | | 44[d] | | 41[d] | | |
| Zinc, mg | 6 | 5.2 ± 0.2 | 71 ± 15 | 5.8 ± 0.6 | 66 ± 9 | 0.35 | 0.62 |
| Vitamin C, mg | 22 | 53 ± 4 | 12 ± 6 | 79 ± 5 | 0 | <0.001 | 0.09 |
| Thiamine, mg | 0.5 | 0.60 ± 0.02 | 19 ± 63 | 0.70 ± 0.06 | 29 ± 14 | 0.12 | 0.42 |
| Riboflavin, mg | 0.5 | 0.68 ± 0.04 | 33 ± 7 | 0.64 ± 0.04 | 31 ± 10 | 0.49 | 1 |
| Niacin, mg | 6 | 8.2 ± 0.4 | 28 ± 9 | 8.9 ± 0.8 | 23 ± 15 | 0.44 | 0.8 |
| Vitamin B6, mg | 0.5 | 1.2 ± 0.05 | 2 ± 4 | 1.0 ± 0.1 | 2 ± 5 | 0.08 | 1 |
| Folate, µg | 160 | 375 ± 29 | 8 ± 26 | 398 ± 43 | 4 ± 12 | 0.66 | 0.66 |
| Vitamin B$_{12}$, µg | 1 | 6.0 ± 1.8 | 16[e] | 1.1 ± 0.2[f] | NE[g] | 0.009 | -- |
| Calcium, mg | 800 | 411 ± 27 | 96 ± 4 | 525 ± 57 | 85 ± 11 | 0.08 | 0.17 |

[a]Intake data are presented as mean ± SEM and represent the mean of a single day's intake for each woman; the SEM refers to inter-individual variation.

[b]Prevalence of inadequacy was estimated as the percent of the usual intake distribution below the estimated average requirement (EAR) for each micronutrient (IOM, 2000) except for iron. For iron, the full probability approach at 7.5% and 10% bioavailability was used (Gibson and Ferguson, 2008). Estimated average requirement for zinc was calculated using physiologic requirements reported by EFSA (EFSA, 2014) and bioavailability assumptions for unrefined cereal diets (IZINCG, 2004). The Institute of Medicine's estimated average requirements for all other nutrients were used. Standard errors (SE) are estimated and reported by IMAPP for the cut-point approach and are reported here.

[c]As retinol activity equivalents (RAE). 1 RAE = 1µg retinol, 12 µg β-carotene, 24 µg α-carotene, or 24 µg β-cryptoxanthin. The RAE for dietary provitamin A carotenoids is two-fold greater than retinol equivalents (RE), whereas the RAE for preformed vitamin A is the same as RE.

[d]Robust SEM could not be calculated as the probability approach rather than the EAR cut-point method was used.

[e]PC-Side was used to estimate B$_{12}$ because it appeared episodically in the diet; the software does not report SEs for the prevalence of inadequacy estimates.

[f]Usual intakes could not be computed, thus arithmetic mean ± SEM are reported.

[g]Not estimable. Distribution was highly skewed.

those in Kongo Central. Food level data revealed greater consumption of animal source foods, specifically fish and beef, among women and children in South Kivu compared to Kongo Central, however the difference was as not as marked (Tables 6 and 7). One of the reasons for this

**Table 6. Top contributing food sources to iron, zinc, and vitamin A intake in WRA.**

| | South Kivu | | | Kongo Central | | |
| Food source | Iron (%) | Zinc (%) | Vitamin A (%) | Iron (%) | Zinc (%) | Vitamin A (%) |
|---|---|---|---|---|---|---|
| Cassava flour | 34 | 27 | -- | 36 | 29 | -- |
| Beans | 20 | 20 | -- | 15 | 15 | -- |
| Amaranth/bean/cassava leaves | 11 | 9 | 16 | 19 | 14 | 19 |
| Sesame | -- | -- | -- | 11 | 12 | -- |
| Fish | 11 | 10 | -- | -- | 6 | -- |
| Beef | -- | 6 | -- | -- | -- | -- |
| White sweet potato | -- | 6 | -- | -- | -- | -- |
| Red palm oil | -- | -- | 70 | -- | -- | 72 |
| Other sources (<5%) | 24 | 22 | 15 | 19 | 24 | 9 |

**Table 7. Top contributing food sources to iron, zinc, and vitamin A intake in children.**

| Food source | South Kivu | | | Kongo Central | | |
|---|---|---|---|---|---|---|
| | Iron (%) | Zinc (%) | Vitamin A (%) | Iron (%) | Zinc (%) | Vitamin A (%) |
| Cassava flour | 27 | 21 | -- | 28 | 23 | -- |
| Beans | 21 | 20 | -- | 18 | 18 | -- |
| Amaranth/bean/cassava leaves | 13 | 11 | 16 | 20 | 14 | 18 |
| Sesame | -- | -- | -- | 13 | 13 | -- |
| Fish | 12 | 11 | -- | -- | 8 | -- |
| Beef | -- | 8 | -- | -- | -- | -- |
| Antelope | -- | -- | -- | -- | -- | 6 |
| White sweet potato | -- | 5 | -- | -- | -- | -- |
| Red palm oil | -- | -- | 67 | -- | -- | 70 |
| Other sources (<5%) | 27 | 26 | 17 | 21 | 25 | 6 |

is due to renewed rebel conflict and political instability in North Kivu that had disrupted the province's agricultural production [26]. Because 68% of households in South Kivu depend on market access to North Kivu to meet food needs, particularly staple crops, political instability contributed to increased food and nutrition insecurity in the region. According to a report by the United States Agency for International Development Office of Food for Peace, the proportion of the population in South Kivu categorized as food insecure at the time of our survey was highest in the country at 64%, of which 14% were classified as severely food insecure [27]. Despite this, our results remarkably show nutrient adequacy for most nutrients.

Nutrient intake patterns and prevalence of inadequacy among women and children in the respective provinces were nearly identical. Calcium and zinc intakes in both regions were low and presented as the greatest micronutrient vulnerabilities in these two regions. The prevalence of inadequate calcium intakes ranged from 60 to 95%, depending on the region and age group. Similarly, prevalence of inadequate zinc intakes was high, ranging from 55 to 90%. It is important to note that there is still considerable debate on the physiological requirements for

**Table 8. Mean usual intakes of selected staples among mothers and children.**

| | South Kivu | | | Kongo Central | | |
|---|---|---|---|---|---|---|
| | % Consumers | Usual intake on consumption days, g/d ± SEM | Usual Intake on all days for entire population, g/d ± SEM[d] | % Consumers | Usual intake on consumption days, g/d ± SEM[d] | Usual Intake on all days for entire population, g/d ± SEM[d] |
| Women 15–49 yr | | | | | | |
| [a]Bean | 100% | 243 ± 14 | 93 ± 7 | 86% | 180 ± 17 | 59 ± 11 |
| [b]Cassava | 100% | 1262 ± 50 | 1178 ± 47 | 100% | 1486 ± 72 | 1471 ± 74 |
| [c]Maize | 64% | 83 ± 10 | 20 ± 5 | 30% | 61 ± 16 | 4 ± 1 |
| Children 0.5–5 yr | | | | | | |
| [a]Bean | 100% | 129 ± 3 | 47 ± 2 | 100% | 97 ± 10 | 37 ± 4 |
| [b]Cassava | 100% | 500 ± 33 | 456 ± 30 | 100% | 640 ± 33 | 604 ± 33 |
| [c]Maize | 53% | 28 ± 2 | 7 ± 2 | 31% | 32 ± 5[e] | 2 ± 0.6[e] |

[a]Bean intakes expressed as dry weight.

[b]Cassava intakes expressed as fresh peeled cassava.

[c]Maize intakes expressed as maize flour.

[d]Usual mean intakes were computed, applying a probability of consumption for foods not consumed every day.

[e]Usual intake could not be computed thus arithmetic mean ± SEM is reported.

zinc [23], which has important ramifications on the estimates of prevalence of inadequacy. However, the general conclusions are consistent with those from the biochemical results that show a moderate to high prevalence of zinc deficiency, in which more than half of women and a quarter of children had low serum zinc concentrations [4].

Our dietary assessment of vitamin $B_{12}$, vitamin A, and folate intakes supports the results from the biochemical survey that show little to no evidence of deficiency in these nutrients. Dietary requirements for vitamin $B_{12}$ according to the various lifestages in our sample are quite low (0.7–2 μg) in comparison to iron (3–8.1 mg) or zinc (3.5–11.6 mg). Therefore, consumption of small quanitites of animal source foods such as red meat and fish likely explains the observed $B_{12}$ sufficiency but not iron or zinc. Food level data revealed vitamin A sufficiency was primarily due to the ubiquitous presence of red palm oil in these regions. Folate intake on the other hand appeared to be moderately inadequate among women at ~20%, and mild among children at ~5% among children. Although prevalence of inadequacy is slightly higher, the results are reasonably consistent with biochemical findings suggesting this is not a problem nutrient.

Our results also show that approximately 30% of women and 20 to 25% of children 1y and older in our sample had inadequate iron intakes, based on a 10% bioavailability assumption. According to our findings from biochemical measures of haemoglobin, ferritin, and soluble transferrin receptor (sTfR), we found little evidence of iron deficiency anemia among women regardless of the biomarker used [4]. In fact, less than 3% of anemia observed was due to iron deficiency. Among children, iron deficiency anemia was higher but remained below 20%. The rates of iron deficiency based on sTfR (sTfR >8.3 mg/L) more closely align with dietary intake results. However, we recently showed in this population that sTfR is a poor diagnostic indicator of iron deficiency in population due to the high prevelance of inherited blood disorders [7]. Indeed, over half of the children in this sample had inherited blood disorders, and the presence of these genetic variants were significantly associated with elevated sTfR concentrations but not ferrtitin or haemoglobin. Thus, ferritin is a better biomarker of iron status in this population. The discrepancy in dietary assessment and biochemical results has been reported in other surveys. Similar results were found in a study conducted by Verbowski et al where dietary intake data contradicted biochemical indicators [28]. In this study, iron deficiency anemia was only attributable to 1.5% of women in their sample, while prevalence of inadequacy was estimated to be 50%. As was highlighted by Verbowski et al, the reason for the discrepancy may be due to the use of bioavailability values for iron that are not suitable for the Congolese diet where starchy roots and tubers contribute roughly 40% of dietary energy intake. For example, for women in Kongo Central, if a 10% bioavailability for iron is assumed 29% of women have inadequate iron intakes but if 7.5% is used this increase to 52%. The bioavailability of iron ranges between < 7.5% and > 17% depending on source. We do not know the overall bioavailability of iron in the Congolese diet.

As expected, cassava is a major staple in the daily diets of the rural Congolese in South Kivu and Kongo Central. Our study showed that women consumed more than 1200 g/d and children more than 500 g/d of cassava thus holding great potential for a cassava biofortification program. Currently, the biofortified cassava program conventionally breeds for pVACS with breeding targets set for 15 μg/g. A recent retention study showed great losses of pVACs during processing and cooking of two primary forms of cassava consumed, *fufu* and *chikwangwe*, with a mere ~ 2% and ~ 10% of pVACS remaining, respectively [29]. These large losses in pVACs are in part a result of the large losses of solid mass due to the processing, with primary cassava products *fufu* and *chikwangwe* containing about one-quarter to one-third of solid mass of the cassava cubes after processing. Despite these losses, our study indicates that a biofortified cassava program can still have important contributions to vitamin A intakes in the

population given the substantial amount of cassava consumed on a daily basis. If only 5% of pVACs are retained, children consuming 500 g/d and women 1000 g/d will still receive an additional 30 to 40% of the average vitamin A requirement from a biofortification cassava intervention. While our results show no evidence of vitamin A inadequacy in the diets of women and children, more than two-thirds of the vitamin A in the diet came from red palm oil. Any disruption to this food source, such as vegetable oil refining, supply chain breaks, or price increases, could cause a dramatic reduction in population vitamin A intakes. Thus, additional dietary sources of vitamin A including carotenoid enriched cassava would be an important safety net.

Beans were also an important part of the diet in both regions, although consumed more often and in greater amounts in South Kivu. Usual intakes of 93 g/d among women and 47 g/d among children would provide ~ 20 to 25% of the iron requirement for women and young children, respectively. Our results on dietary inadequacy show that the age group in greatest need of an iron intervention are infants <1y, but they are also the least likely to benefit from an iron bean biofortification program due to the small amount of food infants consume and their higher iron requirements. Instead, beans may be a more important vehicle for increasing the zinc density in the diets of the women and children in the DRC, which we show is one of the nutrients most lacking in the diet. Indeed, an added benefit to conventionally breeding for iron is that zinc concentrations also increase [30]. Achievements have been made in breeding for low phytic acid content in beans, increasing the potential for zinc absorption [31, 32]. However, zinc absorption studies of low phytic acid beans are required to prove this concept.

In contrast, maize was consumed infrequently and in small quantities during the time of this survey. This could be, in part, a seasonal effect as the harvest season of maize in the eastern provinces was in January and February [26] Production patterns may also explain the low maize consumption, particularly in Kongo Central where maize production is nominal in comparison to cassava, rice and beans [26] Maize, however, is one of three staple crops that are produced above subsistence levels, and active production chains exist, many of which cluster in the southern territories of the DRC [33] Thus, it is also quite possible we did not survey populations in the DRC in which maize constitutes a large share of the diet.

There are a number of strengths in our study, namely that the results from our dietary assessment survey are consistent with the biochemical survey except for iron. Thus, we can draw conclusions with confidence that zinc deficiency prevails as one of the most important micronutrient deficiencies affecting women and children in South Kivu and Kongo Central. Furthermore, there does not appear to be under-reporting of dietary intakes, which frequently occurs with self-reported dietary surveys, as energy intakes were reasonable. Finally, we developed a locally specific food composition table that accurately reflects the Congolese diet giving us further confidence in our results.

There are also important considerations and limitations to keep in mind when interpreting these results. The first is that the timing of the dietary assessment occurred during a period of very high political instability. Thus, the generalizability should be interpreted with some amount of caution. In addition, we did not measure breastmilk intakes among children, which constitutes an important supply of energy and micronutrients in their diets. To account for this limitation, we applied WHO estimates of average breastmilk intakes among children in low and middle income countries, however we have little knowledge if these averages reasonably represent breastmilk intake among children in the DRC region or if they are an over or underestimate. Finally, estimation of probability of consumption and quantity of staple crops requires a minimum two days, thus the data used to conduct these analyses was limited. However, mean usual intakes of all crops reported are in agreement with the arithmetic mean from the full sample, thus giving confidence to our results.

## Conclusions

Dietary data here corroborate the low prevalence of biochemical vitamin A deficiency but not iron. We found respondents in both provinces to be replete with Vitamin A, and this was largely due to a single dietary source, red palm oil. Carotenoid enriched cassava would be a prudent safety net measure to ensure a sustainable source of vitamin A in these provinces of the DRC, in addition to red palm oil and routine supplementation programs. Iron biofortified beans may be warranted given the high rates of dietary inadequacy and uncertainty around the validity of iron biomarkers. There was, however, a high burden of zinc inadequacy ranging from and 55–90%. The high zinc burden was confirmed by biochemical data where more than 50% of women and 25% of children had low serum zinc concentrations. Crops biofortified with zinc therefore appear warranted. We caution that our findings cannot be extrapolated to the entire Congo where diverse agro-ecological landscape exist or when political and environmental challenges and shocks occur which challenge food production.

## Acknowledgments

We are indebted to all the women and children who generously gave their time to participate in our study. We also give special thanks to all the field staff that helped support data collection efforts.

## Author Contributions

**Conceptualization:** Mourad Moursi, Jennifer Foley, Meaghan D. Hawes, Esto Bahizire, Erick Boy, Tim J. Green.

**Data curation:** Kossiwavi Améwono Ayassou Sawadogo, Jennifer Foley, Meaghan D. Hawes, Pierrot L. Tugirimana, Esto Bahizire, Pierre Z. Akilimali.

**Formal analysis:** Moira Donahue Angel, Crystal D. Karakochuk, Thomas R. Sullivan.

**Funding acquisition:** Tim J. Green.

**Investigation:** Mourad Moursi, Jennifer Foley, Esto Bahizire, Erick Boy, Tim J. Green.

**Methodology:** Kristina D. Michaux, Mourad Moursi, Jennifer Foley, Meaghan D. Hawes, Kyly C. Whitfield, Esto Bahizire, Erick Boy, Tim J. Green.

**Project administration:** Mourad Moursi, Kossiwavi Améwono Ayassou Sawadogo, Jennifer Foley, Meaghan D. Hawes, Pierrot L. Tugirimana, Esto Bahizire, Pierre Z. Akilimali, Tim J. Green.

**Supervision:** Mourad Moursi, Tim J. Green.

**Writing – original draft:** Najma A. Moumin, Moira Donahue Angel, Tim J. Green.

**Writing – review & editing:** Najma A. Moumin, Moira Donahue Angel, Crystal D. Karakochuk, Kristina D. Michaux, Mourad Moursi, Kossiwavi Améwono Ayassou Sawadogo, Jennifer Foley, Meaghan D. Hawes, Kyly C. Whitfield, Pierrot L. Tugirimana, Esto Bahizire, Pierre Z. Akilimali, Erick Boy, Thomas R. Sullivan, Tim J. Green.

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
