## [Decision Letter · Decision Letter 0]

28 Oct 2019

PONE-D-19-25185

Micronutrient intake and prevalence of micronutrient inadequacy among women (15-49 y) and children (6-59 mo) in South Kivu and Kongo Central, Democratic Republic of Congo (DRC)

PLOS ONE

Dear Dr Green,

Thank you for submitting your manuscript to PLOS ONE. After careful consideration, we feel that it has merit but does not fully meet PLOS ONE’s publication criteria as it currently stands. Therefore, we invite you to submit a revised version of the manuscript that addresses the points raised during the review process.

The reviewers have provided some clear guidance as to the changes that are needed. Please pay special attention to each of their suggestions, especially the one regarding the need for statistical comparisons.

We would appreciate receiving your revised manuscript by Dec 12 2019 11:59PM. To enhance the reproducibility of your results, we recommend that if applicable you deposit your laboratory protocols in protocols.io, where a protocol can be assigned its own identifier (DOI) such that it can be cited independently in the future. For instructions see: http://journals.plos.org/plosone/s/submission-guidelines#loc-laboratory-protocols

We look forward to receiving your revised manuscript.

Kind regards,

Laura E. Murray-Kolb

Academic Editor

PLOS ONE

Journal Requirements:

1. 

2.  Please include additional information regarding the survey or questionnaire used in the study and ensure that you have provided sufficient details that others could replicate the analyses. For instance, if you developed a questionnaire as part of this study and it is not under a copyright more restrictive than CC-BY, please include a copy, in both the original language and English, as Supporting Information. Moreover, please include more details on how the questionnaire was pre-tested, and whether it was validated.

3.  Please consider including more details on the qualitative methods used during the focus group meetings (for  qualitative research, we suggest consulting the COREQ guidelines: http://intqhc.oxfordjournals.org/content/19/6/349.

Reviewers' comments:

Reviewer's Responses to Questions

**Comments to the Author**

1. Is the manuscript technically sound, and do the data support the conclusions?

Reviewer #1: Partly

Reviewer #2: Yes

2. Has the statistical analysis been performed appropriately and rigorously? 

Reviewer #1: No

Reviewer #2: Yes

3. Have the authors made all data underlying the findings in their manuscript fully available?

Reviewer #1: No

Reviewer #2: Yes

4. Is the manuscript presented in an intelligible fashion and written in standard English?

Reviewer #1: Yes

Reviewer #2: Yes

5. Review Comments to the Author

Reviewer #1: The objective of this study was to determine the micronutrient intakes and prevalence of inadequate intakes in two provinces of the Democratic Republic of the Congo. My main recommendation is that statistical comparisons should be made so that statements can be made about differences in intake. It seems that differences between provinces and age groups would be appropriate. Secondly, the authors should comment on the discrepancy of iron deficiency estimates based on the biochemical data. In the abstract and throughout the manuscript, the authors indicate that there is a low prevalence of iron deficiency in women of reproductive age and their children (~5% and ~3% in women of reproductive age and children, respectively). However, this statement is based on ferritin concentrations that are reported in the primary study (PMID: 26901219). sTfR was also measured in the primary study and using sTfR, the prevalence of iron deficiency in these 2 populations was ~20% and ~63%, respectively, which would be more in line with the dietary data. Additional suggestions are included below:

• The following phrases in the abstract may be unclear to the reader without additional context and should be removed: “(with repeats)”, “(10% bioavailability)”, and “However, any change to the supply of red palm oil would dramatically reduce population vitamin A intakes..”

• Line 45-46: include population. WRA?

• Line 48: A percentage is given for WRA, but not children.

• Is it safe to say that iron biofortified beans may be warranted based on dietary intake data alone? This statement conflicts with the low rates of iron deficiency based on ferritin concentration.

• Citations are missing and should be included (e.g., line 65-66, 79-80).

• Line 71-78: Is this data from the parent study? If so, it should be cited.

• Be consistent with standard deviations and standard errors in the tables.

• Anytime comparisons are made, statistics should be performed (e.g., line 247-248).

• Statistics should be performed in order to compare intakes across provinces, and possible age groups.

• If prevalence of micronutrients besides iron, zinc, and vitamin A are shown in Tables 2-5, should food sources of these nutrients also be shown?

• Is it accurate to say that the results from the dietary assessment survey “mimic” the findings from the biochemical survey?

• Per PLOS ONE policy, a comment needs to be made regarding data availability.

Reviewer #2: The study investigated the prevalence of inadequate intakes of selected nutrients in two provinces of DRC to explain findings from biochemical assessment and justify need for proposed dietary interventions. The manuscript is well written, and the following comments are suggestions to improve the content.

Abstract: methods have not been adequately described in the abstract. Authors should provide a brief description of how the prevalence of inadequate intakes was estimated.

Line 166: How does use of the USDA nutrient database affect the nutrient composition of the foods. Did they use raw foods, or did they also include minimally processed food components also. Some items in the USDA nutrient database may be fortified with vitamins and minerals. Authors should indicate approaches they used to ensure the best choices of the foods from the USDA database were selected to minimize over or underestimation of nutrient contents.

Line 250: 32 % is the average for all subjects. The current presentation suggests that each of the two districts had a prevalence of inadequacy of 32 %. I suggest the authors delete “32 %”.

Table 6: Antelope does not seem to be providing any of the nutrients, yet it is listed among the top sources of the three micronutrients. Or may be the value is missing.

Line 388-390. I wonder if the use of the EFSA physiological requirements for zinc played a role in the very high prevalence of inadequate zinc intakes in the populations studied. How would the results compare if the authors had used the US Institute of Medicine (IOM) physiological requirement values instead, since for all the other analysis they used EAR values from US IOM? Could they have used the IOM physiological requirements and applied appropriate bioavailability estimates and compare results?

Line 396: Animal source foods are rich sources of Vitamin B12, zinc and iron. While vitamin B12 is primarily from animal source foods and fortified foods, plant foods also provide zinc and iron. It is therefore unexpected to find a huge prevalence of inadequate zinc and iron intakes and high level of deficiencies of these two nutrients while vitamin B12 intake is highly sufficient. The authors need to address this discrepancy.

6. PLOS authors have the option to publish the peer review history of their article (what does this mean?). If published, this will include your full peer review and any attached files.

Reviewer #1: No

Reviewer #2: No

---

## [Author Response · Author response to Decision Letter 0]

11 Dec 2019

Response to reviewers has been uploaded as a separate word document.

---

## [Decision Letter · Decision Letter 1]

2 Jan 2020

PONE-D-19-25185R1

Micronutrient intake and prevalence of micronutrient inadequacy among women (15-49 y) and children (6-59 mo) in South Kivu and Kongo Central, Democratic Republic of the Congo (DRC)

PLOS ONE

Dear Dr Green,

Thank you for submitting your manuscript to PLOS ONE. After careful consideration, we feel that it has merit but does not fully meet PLOS ONE’s publication criteria as it currently stands. Therefore, we invite you to submit a revised version of the manuscript that addresses the points raised during the review process.

As you will see from the reviewers' comments, they feel that you've addressed their prior concerns adequately. However, one reviewer brings up a minor (but good) point about the tables. Once this reviewer's concern regarding tables 2-5 is addressed, the manuscript should be ready for acceptance.

We would appreciate receiving your revised manuscript by Feb 16 2020 11:59PM. To enhance the reproducibility of your results, we recommend that if applicable you deposit your laboratory protocols in protocols.io, where a protocol can be assigned its own identifier (DOI) such that it can be cited independently in the future. For instructions see: http://journals.plos.org/plosone/s/submission-guidelines#loc-laboratory-protocols

We look forward to receiving your revised manuscript.

Kind regards,

Laura E. Murray-Kolb

Academic Editor

PLOS ONE

Reviewers' comments:

Reviewer's Responses to Questions

**Comments to the Author**

1. If the authors have adequately addressed your comments raised in a previous round of review and you feel that this manuscript is now acceptable for publication, you may indicate that here to bypass the “Comments to the Author” section, enter your conflict of interest statement in the “Confidential to Editor” section, and submit your "Accept" recommendation.

Reviewer #1: All comments have been addressed

Reviewer #2: All comments have been addressed

2. Is the manuscript technically sound, and do the data support the conclusions?

Reviewer #1: Yes

Reviewer #2: Yes

3. Has the statistical analysis been performed appropriately and rigorously? 

Reviewer #1: Yes

Reviewer #2: Yes

4. Have the authors made all data underlying the findings in their manuscript fully available?

Reviewer #1: Yes

Reviewer #2: Yes

5. Is the manuscript presented in an intelligible fashion and written in standard English?

Reviewer #1: Yes

Reviewer #2: Yes

6. Review Comments to the Author

Reviewer #1: (No Response)

Reviewer #2: The authors have adequately responded to all previous comments. I only have a minor comment on the reported values in some of the tables as follows:

For tables 2-5, under the prevalence of inadequate intakes, the authors have not reported the SE for some of the values. This should be corrected or the reasons for not being able to estimate these values should be mentioned as part of the table footnotes.

7. PLOS authors have the option to publish the peer review history of their article (what does this mean?). If published, this will include your full peer review and any attached files.

Reviewer #1: No

Reviewer #2: No

---

## [Author Response · Author response to Decision Letter 1]

8 Jan 2020

Reviewer Comment: The authors have adequately responded to all previous comments. I only have a minor comment on the reported values in some of the tables as follows:

For tables 2-5, under the prevalence of inadequate intakes, the authors have not reported the SE for some of the values. This should be corrected or the reasons for not being able to estimate these values should be mentioned as part of the table footnotes.

Rebuttal: We thank the reviewer for this comment. The probability approach was used to estimate prevalence of inadequacy for iron rather than the cut-point method. IMAPP does not provide a SE for probability of inadequacy when using this approach. For Vitamin B12, PCSIDE software was used to estimate prevalence of inadequacy as there were too few replicates and it appeared episodically in the diet. PC-SIDE does not report SE for prevalence of inadequacy estimates. These explanations have been added in the footnotes for tables 2-5.

---

## [Decision Letter · Decision Letter 2]

3 Apr 2020

PONE-D-19-25185R2

Micronutrient intake and prevalence of micronutrient inadequacy among women (15-49 y) and children (6-59 mo) in South Kivu and Kongo Central, Democratic Republic of the Congo (DRC)

PLOS ONE

Dear Dr Green,

Thank you for submitting your manuscript to PLOS ONE. After careful consideration, we feel that it has merit but does not fully meet PLOS ONE’s publication criteria as it currently stands. Therefore, we invite you to submit a revised version of the manuscript that addresses the points raised during the review process.

Dear Authors,

we recommended  this  manuscript for acceptance; however, it seems appropriate to include few sentences in the discussion describing the limitation of selection of subjects without chronic condition. Vitamin  and mineral deficiencies are underlying chronic conditions. It is also clear that selection of people with chronic condition also present challenges.  This discussion will provide another angle on the findings in this manuscript and their interpretation.

We would appreciate receiving your revised manuscript by May 18 2020 11:59PM. To enhance the reproducibility of your results, we recommend that if applicable you deposit your laboratory protocols in protocols.io, where a protocol can be assigned its own identifier (DOI) such that it can be cited independently in the future. For instructions see: http://journals.plos.org/plosone/s/submission-guidelines#loc-laboratory-protocols

We look forward to receiving your revised manuscript.

Kind regards,

Ouliana Ziouzenkova, PhD

Academic Editor

PLOS ONE

Reviewers' comments:

Reviewer's Responses to Questions

**Comments to the Author**

1. If the authors have adequately addressed your comments raised in a previous round of review and you feel that this manuscript is now acceptable for publication, you may indicate that here to bypass the “Comments to the Author” section, enter your conflict of interest statement in the “Confidential to Editor” section, and submit your "Accept" recommendation.

Reviewer #3: (No Response)

2. Is the manuscript technically sound, and do the data support the conclusions?

Reviewer #3: Yes

3. Has the statistical analysis been performed appropriately and rigorously? 

Reviewer #3: Yes

4. Have the authors made all data underlying the findings in their manuscript fully available?

Reviewer #3: Yes

5. Is the manuscript presented in an intelligible fashion and written in standard English?

Reviewer #3: Yes

6. Review Comments to the Author

Reviewer #3: Thank you for the opportunity to review this manuscript. The manuscript reads well and provides findings that support the proposed interventions to compact iron and vitamin A deficiencies in the DRC. Below are comments for the authors to consider:

1. How was having “no serious and health conditions or chronic diseases”, one of the inclusion criteria, operationalized for this study? This information might help explain the minimal evidence of iron deficiency anemia reported from biochemical assessments.

2. The study notes a discrepancy between dietary and biochemical assessment of iron status. They state that this discrepancy might be due to the use of nutrient reference values not suitable for a population with a high consumption of starchy roots and tubers. It will be good to expand on this plausible explanation with information on the mechanism by which they think the intake of starchy roots and tubers affect iron absorption and hence iron status.

7. PLOS authors have the option to publish the peer review history of their article (what does this mean?). If published, this will include your full peer review and any attached files.

Reviewer #3: No

---

## [Author Response · Author response to Decision Letter 2]

28 Apr 2020

Please see our response

1. How was having “no serious and health conditions or chronic diseases”, one of the inclusion criteria, operationalized for this study? This information might help explain the minimal evidence of iron deficiency anemia reported from biochemical assessments.

Response: This is a good point and likely an oversight on our part. Although this was an exclusion criterion and asked as a question for all potential participants nobody was excluded because of this. The mothers were primarily a young population and had not developed chronic disease (i.e. non-communicable disease). None of the young children had developed chronic disease. On the other hand, there was a high prevalence of Malaria especially in Kongo Central, but this was asymptomatic, and nobody was excluded

The following has been added at line 247 “No mother/ child dyad was excluded due to the presence of chronic disease.”

2. The study notes a discrepancy between dietary and biochemical assessment of iron status. They state that this discrepancy might be due to the use of nutrient reference values not suitable for a population with a high consumption of starchy roots and tubers. It will be good to expand on this plausible explanation with information on the mechanism by which they think the intake of starchy roots and tubers affect iron absorption and hence iron status.

Response: Again, this is our mistake we do not to mean suggest that requirements (nutrient reference values) are incorrect but rather the assumptions on the bioavailability of iron may be wrong, in our case we used 10% and 7.5% . Animal based foods are the best source of bioavailable iron and plant sources are not as good. In high income countries we might use a bioavailability percentage of 17% whereas in a country where cereal grains are the main source of iron an iron bioavailability of 5-7% may be more appropriate. The lower the bioavailability the more iron that is required to meet recommendations. For example, if the nutrient reference value is 10 and if we assume a 17% bioavailability less iron is needed in the diet than if the bioavailability is 7%. All we are saying here is we do not know the bioavailability of iron in the Congolese diet and this may be a source of the discrepancy.

We have added the following at line 462

As was highlighted by Verbowski et al, the reason for the discrepancy may be due to the use of bioavailability values for iron that are not suitable for the Congolese diet where starchy roots and tubers contribute roughly 40% of dietary energy intake. For example, for women in Kongo Central, if a 10% bioavailability for iron is assumed 29% of women have inadequate iron intakes but if 7.5% is used these increases to 52%. The bioavailability of iron ranges between <7.5% to > 17% depending on source. We do not know the overall bioavailability of iron in the Congolese diet.

---

## [Editor Report · Decision Letter 3]

4 May 2020

Micronutrient intake and prevalence of micronutrient inadequacy among women (15-49 y) and children (6-59 mo) in South Kivu and Kongo Central, Democratic Republic of the Congo (DRC)

PONE-D-19-25185R3

Dear Dr. Green,

We are pleased to inform you that your manuscript has been judged scientifically suitable for publication and will be formally accepted for publication once it complies with all outstanding technical requirements.

With kind regards,

Ouliana Ziouzenkova, PhD

Academic Editor

PLOS ONE
---

## [Editor Report · Acceptance letter]

23 Jan 2020

PONE-D-19-25185R2 

Micronutrient intake and prevalence of micronutrient inadequacy among women (15-49 y) and children (6-59 mo) in South Kivu and Kongo Central, Democratic Republic of the Congo (DRC) 

Dear Dr. Green:

I am pleased to inform you that your manuscript has been deemed suitable for publication in PLOS ONE. Congratulations! Your manuscript is now with our production department. 

With kind regards,

on behalf of

Dr. Laura E. Murray-Kolb 

Academic Editor

PLOS ONE